# A Study of the Genetic Structure of Hybrid Camels in Kazakhstan

**DOI:** 10.3390/genes14071373

**Published:** 2023-06-28

**Authors:** Makpal Amandykova, Kairat Dossybayev, Aizhan Mussayeva, Naruya Saitou, Zhazira Zhunusbayeva, Bakytzhan Bekmanov

**Affiliations:** 1Faculty of Biology and Biotechnology, Al-Farabi Kazakh National University, Al-Farabi Ave. 71, Almaty 050040, Kazakhstan; makpal_30.01@mail.ru (M.A.); kairat1987_11@mail.ru (K.D.); zhazira.zhunusbayeva@kaznu.edu.kz (Z.Z.); bobekman@rambler.ru (B.B.); 2Laboratory of Genetics and Cytogenetics, Institute of Genetics and Physiology SC MSHE RK, Al-Farabi Ave. 93, Almaty 050060, Kazakhstan; aimus_@mail.ru; 3Saitou Naruya Laboratory, National Institute of Genetics, 1111 Yata, Mishima 411-8540, Shizuoka, Japan

**Keywords:** Kazakhstan hybrid camels, whole genome sequencing, single nucleotide polymorphism, genetic structure analysis, phylogenetics

## Abstract

Camel farming is gaining scientific interest due to its unique agricultural characteristics. Camels are versatile for milk and meat production, wool, racing, transport, and tourism. To use their full potential, it is essential to improve our understanding of the genetic structure of these animals. One-humped and two-humped camels have received detailed genetic descriptions, while there is no such information for their hybrids, which outperform their parent species in several agricultural characteristics. Thus, in this study, for the first time, the whole genome sequencing data (WGS) of five hybrid camels bred in the Almaty region of Kazakhstan are presented in comparison with the WGS data of one-humped, two-humped, and wild camels. A total of 43,552,164 single-nucleotide polymorphisms were found across the studied groups. Further comparison of these SNPs showed the following number of private SNPs among the populations: hybrid camels (3,271,083), wild camels (2,515,591), Bactrians (1,244,694), and dromedaries (531,224). The genetic structure of the studied animals was described, and a phylogenetic tree was built to assess their genetic distance. It was found that the studied hybrids are genetically closer to dromedaries since they were on the close branch of the phylogenetic tree.

## 1. Introduction

Camels have been used since about the 15th century BC as a mode of transportation [1], while their domestication took place somewhere between 3000 and 6000 years ago. The family *Camelidae* appeared around 40 million years ago in the middle Eocene, and the family’s first ancestors appeared in North America [2]. Afterward, they differentiated into Old and New World camelids. Molecular studies estimated that this split happened 11–16 million years ago [3] or 25 million years ago [4,5]. The old-world camelids migrated to the eastern hemisphere and split into two species: *Camelus bactrianus* (two-humped camels) and *Camelus dromedarius* (one-humped camels). The divergence of *Camelus dromedarius* and *Camelus bactrianus* species is estimated to have occurred between 4.4 and 8 million years ago [5]. Wild two-humped camels (*Camelus ferus*) separated from domestic two-humped camels between 0.6 and 1.8 million years ago [6].

One-humped camels inhabit the deserts of Arabia, while the primary habitat of two-humped camels is in Asia. These species can be crossed and produce fertile offspring partly because they have the same sets of chromosomes (2n = 74) [7]. The crossbreeding of one-humped and two-humped camels has been practiced in the Middle East and Central Asia since ancient times to produce camels with a muscular body constitution, which are more enduring for transporting goods in trade caravans and for military use. Camel hybridization is now widely used in two countries: Turkey and Kazakhstan. In Turkey, hybridization is widespread in order to obtain large animals for participation in annual camel fights, whereas in Kazakhstan, hybridization is utilized to obtain more productive camels that can better withstand the harsh climate of Central Asia [8]. According to some authors, hybrid wool productivity increased by 192% compared with the pure dromedary breed, and hybrid milk production was 69% higher than the Bactrian and even slightly higher (2%) than the dromedary. The milk composition differed between the two species, with the hybrids being intermediate: Bactrian milk had a higher concentration of vitamin C and minerals and a higher fat content but lower long-chain fatty acids [9,10]. Furthermore, due to the heterosis, hybrids have advantages over their parents in terms of precocity and high productivity [11].

Even though the genetic status of the two domestic species has long been established, wild Bactrian camels were only recently recognized as separate species based on molecular genetic data, particularly mitochondrial DNA and nuclear markers [6,12]. In 2002, they were classified as critically endangered animals; there were approximately 950 wild Bactrians worldwide. In May 2006, wild Bactrians were listed in the IUCN Red List of Threatened Species. Currently, there are only three separate habitats for wild camels; 600 individuals are based in northwestern China (Lake Lop, the Taklamakan desert, and the Arjin Shan ranges) and 450 individuals are based in southwestern Mongolia (Trans-Altai Gobi desert). It is worth noting that wild camels are unique objects for studying the adaptive mechanisms of the immune system, as they have adapted to live in the harsh conditions of a desert climate [13]. Research on discovering the essential genes involved in their adaptation to the desert environment may lead to understanding the disease resistance in various animal species, which can be used in breeding programs.

In Kazakhstan, both Bactrians and dromedaries and their hybrids are widespread; in some cases, all three representatives can be found on the same farm. Camel breeding is one of the most profitable sectors of the Kazakhstan economy, as it makes it possible to fully meet the population’s needs with regards to wool, milk, and meat in the country’s desert climate. In 2022, the camel population in Kazakhstan reached 2.6 million [14]. The zones with the most significant distribution of camels in Kazakhstan include the Aktobe, Mangystau, South Kazakhstan, Kyzylorda, Atyrau, and Almaty regions. Hybridization between one-humped and two-humped camels has various options for crossing. Depending on the initial parental forms, in Kazakhstan, two methods of hybridization have been identified: Kazakh—when a female Bactrian is crossed with a male dromedary—and Turkmen—when a female dromedary is crossed with a male Bactrian. While studying the genetics of the productive profile of Kazakh camel populations, Baimukanov et al. established the genetic profile of camels (live weight, milk production, wool shearing, and body measurements). The study’s results showed the effectiveness of breeding hybrid camel dams of the dromedary group for the production of camel milk due to the optimal ratio of milk fat and protein [11].

The rapid development of molecular genetics methods over the past decades has positively affected the prospect of obtaining extensive data on the genetic structure of camels (genetic variation, molecular markers, parentage control, the gene of interest, whole genome, dating, and others). To date, the genetic study of camels is sorted into several categories, such as the following: the assessment of genetic diversity and population structure for studying local or regional groups; large-scale phylogenetic studies investigating the demographic history of populations; studies investigating the association between phenotypic traits and underlying genotypes for using marker-assisted breeding programs or adaptation to a particular environment [12]. Also, the genetic data provide important clues about the breeding strategies and selection processes of camels that people used in antiquity. Other important methods for studying the genetics of camels are described in detail by Piro et al. [15].

The use of microsatellites is considered to be a powerful tool for tracking alleles in a population and assessing genetic variation within and between breeds [16]. The international panel on Animal Genetic Diversity (ISAG-FAO, 2004) recommended a list of 25 microsatellites markers for the evaluation of genetic diversity in camelids; 16 of them are considered to be the most polymorphic and are thus highly recommended for dromedary camel genetic characterization [15]. The genetic diversity of four groups of Moroccan camels was studied based on these 16 microsatellite markers, which showed high informativeness, revealing a high level of diversification of the studied genetic resources of camels [17]. Furthermore, the different studies on dromedary genetic diversity were not limited to these 16 microsatellites but instead used a range of varying microsatellite markers whose number and name differ from one study to another [15]. Similar results were obtained in a study of two groups of Saudi Arabian dromedary camels, where 21 microsatellite loci were examined, and it was found that very little gene flow occurred between these groups [18].

The mitochondrial DNA (mtDNA) reflects maternal inheritance and helps establish genetic variation between species [15]. The three existing species of camels inhabit hot and low-mountainous deserts, as well as cold and high-altitude deserts. mtDNA sequencing contributes to studying the role of natural selection under different environmental pressures and can improve our understanding of the evolutionary history of this genus. As a result of these studies, the evolutionary time of divergence between domestic and wild Bactrian camels is estimated to be around 1.1 (0.58–1.8) million years ago [6]. On the other hand, mtDNA sequence analysis revealed dromedary domestication processes and the dynamics of domestication and cross-continental movement of the dromedary camel. Almathen et al. combined ancient DNA sequences of wild and early domesticated dromedary samples from arid regions with nuclear microsatellite and mitochondrial genotype information, demonstrating the potential of using mitochondrial DNA for ancient camel studies [19].

SNP chip technology is also considered a highly informative method for studying animals’ genetic structure and genetic diversity. These markers can be beneficial because they are easy to use and reproducible. The development of SNP markers that cover the coding part of the genome is also needed to understand the relationship between genetic and phenotypic variations in camels or other species [15,16].

Whole genome sequencing (WGS) is the most effective technology conducted in animal husbandry, which makes it possible to extensively understand the genomic structure of the studied farm animals. Unlike SNP chip technology, WGS can produce complete structural genetic information [20]. Nuclear SNPs are also applied in some studies of the dromedary camel [21]. These markers can be used as an alternative to microsatellites, especially in genetic diversity studies, detecting relevant phenotypic traits in livestock [15]. For example, when comparing the genetic sequences of one-humped and two-humped camels, 116,000 SNPs were identified, which determined that 85% of the genomic sequences expressed in the one-humped camel can be found in the two-humped camel [22]. These results may provide some insight into the evolution of Bactrian camels, and the discovery of genome-wide genetic variation paves the way for developing techniques such as genome-wide association studies and genomic selection [23]. In another context, Ruiz et al. [24] developed a diagnostic panel of SNPs to identify the hybridization patterns in camels with uncertain origins to support hybrid breeding management and detect potential rare dromedary introgression in the remaining wild Bactrian camels in Mongolia and China. Recently, in a genome-wide association study, by using genotyping via sequencing, Sani et al. [25] were able to identify 99 SNP markers that could be associated with important traits to improve camel breeding, namely, birth, weight, daily gain, and body weight [26].

Other studies also describe the distribution, breeds, evolutionary history, domestication, and genetic diversity of various camel populations based on mtDNA, nuclear DNA, and whole genome sequencing [27,28], such as the identification of the association of polymorphisms in growth hormone, insulin-like growth factor-I, and other genes with some physical characteristics of camels [27]. While most of these studies covered major geographic regions/countries and identified millions of SNPs amenable to implementation within array-based genotyping platforms, future studies will likely focus on in-depth analyses of within-population/within-country genetic diversity to capture fine-scale genomic variation and population structures. The latest technologies for studying the genetic structure, including the genotypes and phenotypes of pure breed Bactrian and dromedary camels and their hybrids, will allow us to build each “breed standard” and detect genotypes with highly productive features [29].

It should be noted that the genetic characteristics of Kazakhstani camel populations are yet to be studied, apart from the studies based on microsatellites [30] and camel casein gene polymorphism [31]. As for hybrids, data on studies of their genetic characteristics still need to be included. Thus, in this study, we aim to identify the genome-wide variations (SNPs) in Kazakhstani hybrids of one-humped and two-humped camels using the whole genome sequencing data of five hybrid camels compared with the whole genome sequencing data of six Bactrian, four dromedary, and five wild camels. Our main goal is to describe the genetic structure of hybrid camels in comparison with their parental species (one-humped and domestic two-humped camels) and wild camels.

## 2. Materials and Methods

### 2.1. Sample Collection

Whole genome sequencing data of 20 camels was used for the study: six samples of *Camelus bactrianus* (Kazakhstan camel population, n = 6♀), four samples of *Camelus dromedarius* (Iranian camel population, n = 3♀; n = 1♂), five samples of *Camelus ferus* (Gobi Desert, Mongolia, the sex of the animals is missing), and five samples of hybrids (n = 5♀) of one-humped and two-humped camels (Kazakhstan). Information on the parental species of hybrid camels is presented in Table 1.

Information about the parental individuals of hybrids often plays an essential role in research on the mitochondrial genome of hybrids. However, it may also be the key to understanding some study results when studying their genetic structure using whole genome sequencing data.

The first 15 samples (the age and developmental stage of the animals are missing) were taken from the NCBI database from a study by Chinese researchers on the whole genome sequencing data of 128 camels (GenBank accession number SRP107089). For the remaining five hybrids, whole genome sequencing was performed. The number of studied animals from each population was determined following the number of sequenced samples of hybrid camels to balance the obtained results. The blood samples of the hybrids aged 2–3 years were collected by a veterinarian in EDTA vacuum tubes on the camel farm in the Akshi village of the Ili district of the Almaty region (Kazakhstan), which is one of the largest camel breeding centers in this area. Camels of both species (Bactrian and dromedary and their hybrids) are bred here. All animal care and experiments were approved by the Local Ethics Committee of the Institute of Genetics and Physiology SC MSHE RK (14 April 2022, Almaty, Kazakhstan). All the samples and data processed in our study were obtained with the consent of the breeding farm. To reduce the stress on animals, positive rewards such as petting were implemented before restraint for blood collection. Further, the blood samples were delivered to the Laboratory of Animal Genetics and Cytogenetics of the Institute of Genetics and Physiology (Almaty, Kazakhstan) in containers with refrigerant and stored in a freezer (at −25 °C) until used for DNA extraction.

### 2.2. DNA Extraction and Quality Assessment

DNA extraction was performed using the GeneJET Genomic DNA Purification Kit (ThermoScientific, Waltham, MA, USA) according to the manufacturer’s protocol. For preparing the extraction solution, 1:1 volume of Binding Buffer was added to the completed PCR mixture and mixed, yielding a yellow-colored solution, which indicated an optimal pH for DNA binding. A total of 800 μL of the yellow-colored solution was transferred up to the GeneJET. The flow-through purification column was discarded after centrifuging for 30–60 s. Then, 700 μL of Wash Buffer was added to the GeneJET purification column and centrifuged for 30–60 s. After discarding the flow-through, the purification column was placed back into the collection tube. A total of 50 μL of Elution Buffer was added to the center of the GeneJET purification column membrane and centrifuged for 1 min. The purified DNA was stored at −20 °C prior to use.

The determination of the quantitative and qualitative indicators of DNA was carried out on a NanoDrop One and a Qubit Fluorometer (ThermoScientific, USA). The obtained data are presented in Table 2.

Whole genome sequencing requires DNA at a concentration of at least 50 ng/µL. Thus, the quantitative indicators of our DNA samples were more than suitable for further analysis. DNA quality was also checked via agarose gel electrophoresis.

### 2.3. Whole Genome Sequencing

Sequencing was carried out on the Illumina New Generation Sequencing platform (NovaSeq 6000) from Macrogen company (Seoul, Republic of Korea) and consisted of four stages: quality control of DNA samples and preparation for the library construction; library construction; sequencing; acquiring raw data in *.fastq file format. The sequencing library was prepared via the random fragmentation of the DNA samples, followed by 5′ and 3′ adapter ligation. Alternatively, “tagmentation” combines the fragmentation and ligation reactions into a single step, which significantly increases the efficiency of the library preparation process. Adapter-ligated fragments were then PCR-amplified and gel-purified. The library was loaded into a flow cell for cluster generation, where fragments were captured on a lawn of surface-bound oligos complementary to the library adapters. Each fragment was then amplified into distinct, clonal clusters through bridge amplification. Illumina SBS technology utilizes a proprietary reversible terminator-based method that detects single bases as they are incorporated into DNA template strands. As all four reversible, terminator-bound dNTPs were present during each sequencing cycle, natural competition minimized incorporation bias and significantly reduced raw error rates compared to other technologies, resulting in highly accurate base-by-base sequencing, which virtually eliminates sequence-context-specific errors, even within repetitive sequence regions and homopolymers. Sequencing data were converted into raw data for analysis. The Illumina sequencer generates raw images utilizing sequencing control software for system control and base calling through an integrated primary analysis software called RTA (Real Time Analysis). The BCL (base calls) binary was converted into FASTQ utilizing the Illumina package bcl2fastq. The following indicators were calculated for the raw data: the total number of bases; the total number of reads; GC content (%); quality score of 20 (Q20, %), which represents an error rate of 1 in 100 (meaning every 100 bp sequencing read may contain an error), with a corresponding call accuracy of 99%; and quality score of 30 (Q30, %) with a call accuracy of 99.9%.

### 2.4. Sequencing Data Analysis

The processing and analysis of raw sequencing data were performed in several stages using bioinformatics analysis methods. Quality control of the raw sequencing reads was performed using FastQC v.0.12.0 (http://opengene.org/fastp/fastp/, accessed on 11 June 2023), which provides an overview of basic quality control metrics for raw sequencing data [32]. Clean reads were mapped to the *Camelus ferus* reference assembly (NCBI Accession: NW_006210177.1) using Burrows–Wheeler aligner v.0.7.15 (BWA-MEM algorithm) (https://github.com/bwa-mem2/bwa-mem2/, accessed on 11 June 2023), which has the best performance based on alignment rate and gene coverage [33]. The SAMTools v.1.17 (https://github.com/samtools/, accessed on 11 June 2023) [34] was used to sort and index the files. Further, the identification of repeated reads was carried out on the Picard tool v.3.0.0 (https://github.com/broadinstitute/picard/, accessed on 11 June 2023) [35], and the identification of differences between aligned reads and the reference genome was carried out on bcftools v.0.1.13 (https://github.com/samtools/bcftools/, accessed on 11 June 2023) [36]. The quality control of the obtained SNPs was carried out based on Plink v.1.9 (https://www.cog-genomics.org/plink/, accessed on 11 June 2023) [37] to remove SNPs with insufficient genotyping quality using several parameters, such as the following: absence of SNPs in a significant part of the individuals, individuals with a high level of genotype absence (--geno 0.01, --mind 0.02), a low minor allele frequency threshold set to eliminate genotyping errors (--maf 0.01) and Hardy–Weinberg equilibrium (--hwe 0.001), individuals with a high or low heterozygosity index (--indep 50 5 2).

### 2.5. Genetic Structure Analysis

Admixture v.1.3 (https://dalexander.github.io/admixture/, accessed on 11 June 2023), which provides maximum likelihood estimations of individual ancestries from multilocus SNP genotype datasets [38], and PCA (https://principal-component-analysis-pca.apponic.com/, accessed on 11 June 2023), which estimates population structure and sample ancestry in genetic datasets [39], were used for a comparative description of the genetic structure of the studied animals. The R Package v.4.2.3 (Shortstop Beagle) (https://www.r-project.org/, accessed on 11 June 2023) [40] was used to search for breed-specific and breed-wide SNPs. The neighbor-joining (NJ) method, which is based on the distance between species [41], was used to construct the phylogenetic tree, and the results were visualized using the FigTree program v.1.4.4 (http://tree.bio.ed.ac.uk/software/figtree/, accessed on 11 June 2023) [42].

## 3. Results and Discussion

### 3.1. Whole Genome Sequencing and Data Filtering

As a result of sequencing, raw data in *.fastq format was obtained, and the total number of reads varied from 717,006,544 to 1,377,460,994 among five camels. This result was consistent with the whole genome sequencing results of Iranian camels, as the average number of reads is fairly similar [23,43]. The average GC content, which typically includes many protein-coding genes, was 41.92%, and Q30 was 92.99%. Our sequencing data showed a high Phred quality (Q20 > 97%, Q30 > 82%), which is a measure of the quality of the identification of the nucleobases generated by automated DNA sequencing, and the GC content was stable at between 41.84 and 42.01% (Table 3 and Figure 1).

### 3.2. SNP Analyses

A total of 43,552,164 SNPs were identified in the studied animals (20 camels). These data are consistent with the sequencing results of Iranian dromedaries, and this was observed after comparing the number of studied animals and the identified SNPs in both studies [23]. There were differences in the number of SNPs of the studied groups, which were in the following order: Hybrid camels > Dromedary camels > Bactrian camels > Wild camels. In contrast to the study of 128 camels, in our work, the number of identified SNPs in dromedaries (10,872,754) was higher than in Bactrians (9,310,730) [44]. Furthermore, hybrid camels were also included in our research, and we can see that the number of SNPs among hybrids was the highest (15,904,987) among all groups studied. This may be because their genome includes the genomes of both Bactrians and dromedaries. The lower number of identified SNPs in Mongolian wild camels (7,463,693) could be attributed to their higher resemblance to the reference genome (NCBI Accession: NW_006210177.1). While conducting quality control on the SNPs using the Plink v.1.9 program [45], the initial number of SNPs (43,552,164) decreased due to the cut-off of SNPs that did not correspond to the established quality control parameters. Thus, 18,830,429 SNPs for all studied animals were generated. After the first step of quality control, 470,961 SNPs passed the filters. After the exclusion of SNPs absent in a significant part of the individuals and the exclusion of individuals with a high level of genotype absence (--geno 0.01, --mind 0.02), a low minor allele frequency threshold was set to eliminate genotyping errors (--maf 0.01) and Hardy–Weinberg equilibrium (--hwe 0.001), resulting in a total of 253,213 SNPs being selected. Another important step to note is the exclusion of individuals with a high or low heterozygosity index (--indep 50 5 2), which resulted in 31,015 SNPs being selected for further study.

### 3.3. Population Structure Analysis

A genetic analysis of the population structure was performed to estimate the different ancestral proportions among the studied camel groups. Diversity in SNPs across the groups was determined for the various most probable number of populations (K), where the best K value indicates a low cross-validation error (CV) compared to other K values. The CV score found the lowest error level at K numbers 2, 3, 4, and 5 (Figure 1B). The cross-validation procedure supported that K = 2 was optimal, showing a clear division between the populations (Figure 1A).

At a value of K = 2, we noticed that Bactrians and wild camels are almost similar since they descend from the same species and are both two-humped. The exact similarity is observed in dromedaries with hybrids. However, by increasing the value to K = 3 or more, the separation of Bactrians and wild camels from other groups can already be observed. The PCA supported this statement, showing Bactrians and wild camels grouped distantly from other groups (Figure 3). Despite the increase in the K value, the similarity of dromedaries and hybrids persisted and was confirmed by PCA Plot (Figure 2) and our phylogenetic tree (Figure 3), indicating the introgression of dromedaries into the hybrid camels.

### 3.4. Phylogenetic Tree Construction

The phylogenetic tree (Figure 3), constructed by employing the Neighbor Joining method, showed that the wild camels (*Camelus ferus)* and domestic Bactrian camels (*Camelus bactrianus*) had distinct branches that were distant from the other two groups of camels. In contrast, branches of hybrids were close to dromedary camels (*Camelus dromedarius*). Thus, it can be assumed with great certainty that the first generation of hybrids under study was then repeatedly crossed with dromedaries, which is unsurprising. Crossbreeding hybrid camels with each other usually has the opposite effect and leads to a decrease in the productive qualities of animals; therefore, usually, the first generation of camels is crossbred by one of the parental representatives [11]. However, two hybrid camels are still grouped into branches separate from dromedaries. The exception is one dromedary, which was located among hybrids in the phylogenetic tree. This can be explained by the mosaic structure of the genome of this camel, i.e., due to being phenotypical of dromedaries, the genome of this camel includes the genome composition of both one-humped and two-humped camels and indicates the presence of hybridization.

To determine the common and private SNPs found in the studied populations, the frequency of occurrence of alleles was determined, and the Plink program was used to select SNPs with a frequency equal to zero. The result is visually represented in the Venn diagram below (Figure 4).

A comparison of identified SNPs of the studied groups revealed 2,631,403 SNPs (13%) shared across all populations. Interestingly, hybrid camels had the largest number of unique SNPs (3,271,083), pointing to their genetic distance from other studied groups of camels. Additionally, 2,515,591 of the identified SNPs are unique to wild camels, while in Bactrians, this number is 1,244,694, and the fewest private SNPs were found in dromedaries (531,224). Hybrids and dromedaries had the most significant number of common SNPs (22%), which was significantly less than those in hybrids with Bactrians and hybrids with wild camels (4% and 1%, respectively). This also confirms the proximity of hybrids to dromedaries. Additionally, it should be noted that 13% of the identified SNPs were common for hybrids, dromedaries, and Bactrians, which is quite logical, given that dromedaries and Bactrians are the genetic parents of hybrid camels [11]. Wild camels also share more SNPs with Bactrians than with dromedaries, consistent with the results of the 128 camel sequencing data [44]. Since the Bactrians and dromedaries used in this study have different geographical origins, hailing from either Kazakhstan or Iran, future studies should aim to comparatively analyze Bactrians and dromedaries of Kazakhstan origin to check the presence of common SNPs, taking inspiration from earlier research works [23,46].

The results indicated that wild camels had the highest nucleotide diversity and genetic diversity; these features may be attributable to the preservation of the genetic diversity of this group’s ancestors. Population structural analysis results suggested that wild camels were phylogenetically distant from camels of other groups; this may be due to the geographical distribution of the wild camels. Wild camels are common in parts of northwestern China and southwestern Mongolia, which is far from the location of the domestic camels, resulting in less genetic communication between wild and domestic camels. Also, it can be assumed that this separation of wild camels from domestic ones results from the lack of directed artificial selection among their populations, which leads to a low rate of inbreeding. The process of natural selection emphasizes high adaptability to one’s habitat. This result was consistent with a study by Ming et al., wherein the separation of wild camels from other groups of studied camels, including dromedaries; Iranian, Kazakh, and Russian Bactrians; and others, could be seen [44]. Furthermore, another study [43] showed low heterozygosity in wild Bactrian camels, which was explained by their small population size.

Even though hybrid camels were close to dromedaries in terms of genetic structure, the separation between two hybrids and the dromedaries on the phylogenetic tree may provide a reason to assume that hybrid camels form their unique genome differently to their parental species.

## 4. Conclusions

We used the whole genome sequencing data of five hybrid camels and publicly available genomic data of other camels to analyze the SNPs present in the genomes of Bactrian, dromedary, wild, and hybrid camels and systematically studied their genetic diversity and genetic structure. This study revealed differences in the genome of hybrids derived from one-humped, two-humped, and wild camels. Moreover, hybrid camels had the most significant number of unique SNPs compared to parental species. The results of the present study could facilitate further studies into the genetic resources of this mammal, as well as the conservation of these resources. These investigations require subsequent validation so we can better understand the concepts behind these results, which can be achieved by increasing the number of studied animals and performing more studies in this field. At the same time, the results of this study can serve as comparative data for new research on this topic.

## Figures and Tables

**Figure 1 genes-14-01373-f001:**
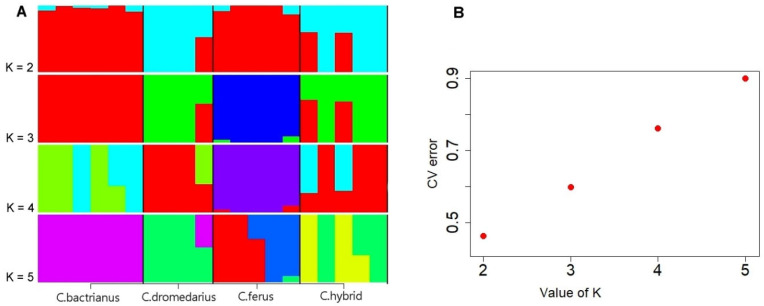
(**A**) ADMIXTURE analysis for *Camelus bactrianus*, *Camelus dromedarius*, *Camelus ferus*, and *Hybrid camel* (K value shown from 2 to 5). (**B**) The CV error of each K value.

**Figure 2 genes-14-01373-f002:**
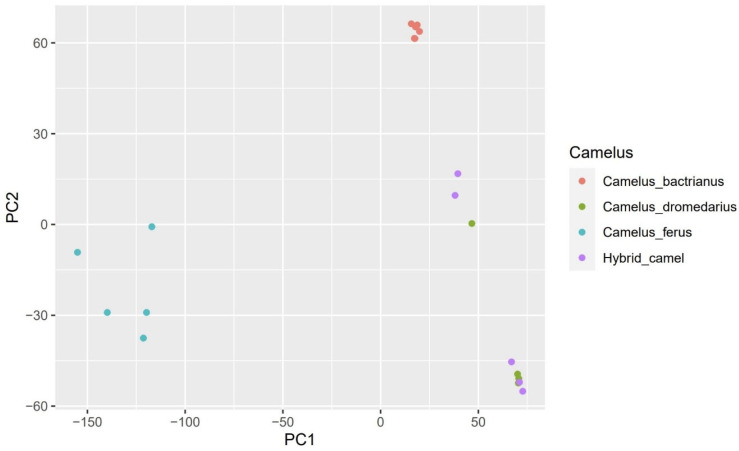
Principal component analysis (PCA) of *Camelus bactrianus*, *Camelus dromedarius*, *Camelus ferus*, and *Hybrid camel* samples based on SNP data. All samples are represented by a dot and are colored according to the type of camel.

**Figure 3 genes-14-01373-f003:**
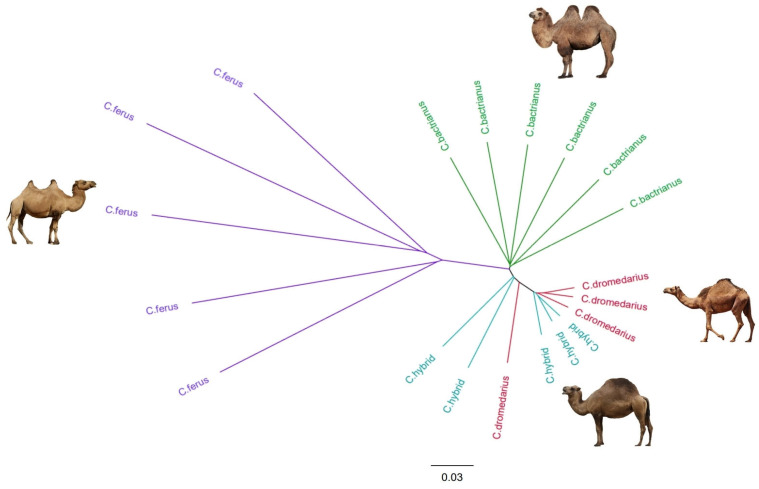
Phylogenetic tree constructed with the neighbor-joining (NJ) method for *Camelus bactrianus*, *Camelus dromedarius*, *Camelus ferus*, and *Hybrid camel*. (Images of camels not to scale and were taken from open sources http://www.pinterest.com/, accessed on 12 May 2023).

**Figure 4 genes-14-01373-f004:**
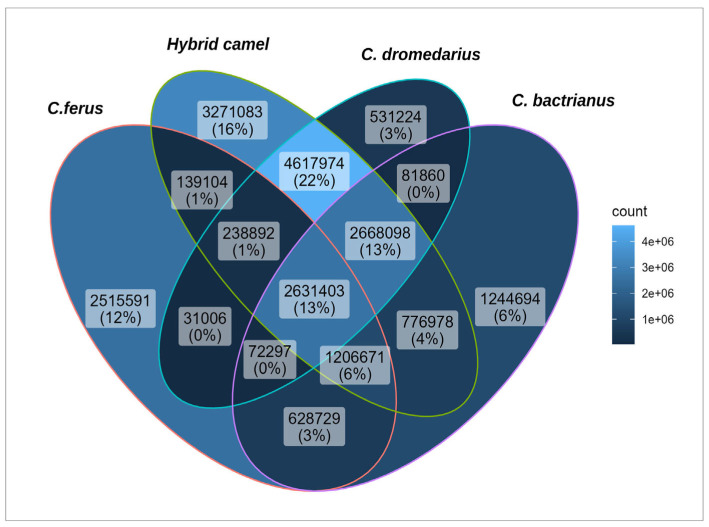
Venn diagram showing the overlap of SNPs for *Camelus bactrianus*, *Camelus dromedarius*, *Camelus ferus*, and *Hybrid camel*.

**Table 1 genes-14-01373-t001:** Hybrid camels origin information.

№	Description
Hybrid-1	♀ Bactrian × ♂ Dromedary
Hybrid-2	♀ Bactrian × ♂ Dromedary
Hybrid-3	♀ Dromedary × ♂ Bactrian
Hybrid-4	♀ Bactrian × ♂ Dromedary
Hybrid-5	♀ Bactrian × ♂ Dromedary

**Table 2 genes-14-01373-t002:** DNA concentration of hybrid camels.

№	DNA Samples	DNA Concentration (ng/µL)	
		NanoDrop One Data	Qubit Fluorometer Data
1	Hybrid-1	57.6	53.4
2	Hybrid-2	67.8	60.4
3	Hybrid-3	51.2	48.6
4	Hybrid-4	69.1	62.1
5	Hybrid-5	55.3	53.6

**Table 3 genes-14-01373-t003:** Raw data statistics.

Sample ID ^1^	Total Read Bases (bp) ^2^	Total Reads ^3^	GC (%) ^4^	AT (%) ^5^	Q20 (%) ^6^	Q30 (%) ^7^
Hybrid camel-1	129,838,121,006	859,855,106	41.94	58.06	97.17	92.70
Hybrid camel-2	137,538,079,710	910,848,210	41.92	58.08	97.32	93.02
Hybrid camel-3	207,996,610,094	1,377,460,994	41.91	58.09	97.23	82.89
Hybrid camel-4	108,267,988,144	717,006,544	42.01	57.99	97.38	93.22
Hybrid camel-5	134,823,736,024	892,872,424	41.84	58.16	97.36	93.13

^1^ Sample name. ^2^ Total number of bases sequenced. ^3^ Total number of reads. For Illumina paired-end sequencing, this value refers to the sum of read 1 and read 2. ^4^ GC content. ^5^ AT content. ^6^ Ratio of bases that have a Phred quality score of over 20. ^7^ Ratio of bases that have a Phred quality score of over 30.

## Data Availability

The whole genome sequencing raw data of the hybrid camels used in this study are available from the GenBank BioProject number PRJNA985411.

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
