# Peer review of "A Study of the Genetic Structure of Hybrid Camels in Kazakhstan"

_genes, 2023, doi:10.3390/genes14071373_

Round 1

Reviewer 1 Report

This is an interesting and well-written manuscript aimed to study the genomic structure of hybrid camels and to compare with one-humped, two-humped and wild camels. The introduction provides interesting information. Results are clear and discussion is well described and supported with several references. However, a more detailed description of some molecular processes should be useful for a better understanding of the methods employed in the study. I also suggest considering some minor comments:

-       Line 28: Please check indentation.

-       Line 66: Replace “is” by “includes”.

-       Line 76: Replace “Piro” by “Piro et al.”.

-       Line 77: Replace “are” by “is”.

-       Line 151: The unit of measure should be corrected as follows: ºC.

-       Lines 154-157: I suggest explaining briefly the DNA extraction procedure and indicate which were the quantitative and qualitative indicators for the DNA.

-       Lines 159-163: I suggest explaining briefly the process for whole genome sequencing.

-       Lines 170-175: I suggest mentioning the webpage for each bioinformatics tool.

-       Lines 172-173: Parameter for SNPs quality control should be established in this section.

-       Lines 173-177: I suggest separating this paragraph in a further subheading (i.e., Genetic structure analysis).

-       Line 296: Replace the colon sign by a semicolon.

-       Line 303: Remove the word “And”.

-       Line 307: Move the reference number to the end of the sentence.

-       Lines 314-317: This sentence is not a conclusion, I suggest removing it.

Reviewer 2 Report

Suggestions,

1)    Fig. 1. You already have this information in Table 2. You can remove this.

2)    Did author perform the sequencing of data in this study? If so, please submit the raw sequencing data to NCBI-SRA.

3)    Line 156. How sequencing libraries prepared? The information is missing.

4)    Line 168. Why BWA aligner? Why not hisat2, tophat or STAR?

5)    Line 174. Which tool in R?

6)    Line 179. Where are DNA quality results?

7)    How many reads (percentage) mapped to genome? 

8)    Please consider rewriting the whole manuscript. Many sentences should be reworded or paraphrased. (Important)

9)    Extensive editing of English language required.

Extensive editing of English language required.

Round 2

Reviewer 2 Report

Please consider submitting the raw sequencing reads to NCBI database, its mandatory step before publications. Otherwise same data can be used for multiple publications. 

All other comments has been addressed well. 

Minor editing of English language required
